# Review of Therapeutic Applications of Radiolabeled Functional Nanomaterials

**DOI:** 10.3390/ijms20092323

**Published:** 2019-05-10

**Authors:** Jongho Jeon

**Affiliations:** Department of Applied Chemistry, School of Applied Chemical Engineering, Kyungpook National University, Daegu 41566, Korea; jeonj@knu.ac.kr; Tel.: +82-53-950-5584

**Keywords:** nanomaterial, radioisotope, radionuclide therapy, α-particle, β-particle, radiolabeling, cancer

## Abstract

In the last two decades, various nanomaterials have attracted increasing attention in medical science owing to their unique physical and chemical characteristics. Incorporating radionuclides into conventionally used nanomaterials can confer useful additional properties compared to the original material. Therefore, various radionuclides have been used to synthesize functional nanomaterials for biomedical applications. In particular, several α- or β-emitter-labeled organic and inorganic nanoparticles have been extensively investigated for efficient and targeted cancer treatment. This article reviews recent progress in cancer therapy using radiolabeled nanomaterials including inorganic, polymeric, and carbon-based materials and liposomes. We first provide an overview of radiolabeling methods for preparing anticancer agents that have been investigated recently in preclinical studies. Next, we discuss the therapeutic applications and effectiveness of α- or β-emitter-incorporated nanomaterials in animal models and the emerging possibilities of these nanomaterials in cancer therapy.

## 1. Introduction

Recent advances in nanotechnology and materials science have inspired the development of a wide range of organic and inorganic nanomaterials for applications in preclinical studies and clinical trials. The unique characteristics of nanomaterials can enable targeted drug delivery, accurate diagnosis, and effective treatment of diseases such as cancers [1,2,3,4,5,6,7]. Radioisotopes are a crucial component in nanomedicine. Radiolabeling is a well-established and useful technique for quantitative in vivo assessment of the biological uptake and pharmacokinetics of synthetic nanomaterials [8,9,10,11]. Several γ-ray-emitting radionuclides including positron emitters (β^+^ decay) have been extensively used for developing nanomaterial-based diagnostic agents for positron emission tomography (PET) or single-photon emission computed tomography (SPECT) [12,13,14,15]. These radiolabeled materials can be used to visualize tumor tissues in living subjects as well as other important biological phenomena. In recent years, therapeutic radionuclides (α- and β-emitters) have also been used in clinical applications, and some of these trials have shown significant impacts on tumor treatment [16,17,18,19,20]. Therefore, there is increasing interest in using a combination of therapeutic radioisotopes and nanosized materials for developing promising candidates for new radiopharmaceuticals. Radioisotope-incorporated nanomaterials show favorable properties in vivo compared to bare radionuclides. One major advantage of radiolabeled nanomaterials is their potential ability to contain multivalent radioactive elements in a single carrier [21,22,23]. In general, only one or few radionuclides can be labeled with a typically used biomolecule (e.g., antibodies, peptides) or small-molecule drug. However, multivalent incorporation of radionuclides in a nanoparticle enables transporting numerous α- or β-emitters to cancer cells. Moreover, nanomaterials can be designed to conjugate with various functional molecules such as chemotherapeutic drugs, contrast agents, or cancer-targeting molecules (e.g., antibodies, peptides, and small-molecule ligands) [24,25,26]. This strategy allows the preparation of versatile functional probes that are highly useful in preclinical research including combination therapy or theranostic applications (Figure 1). Another advantage is that the enhanced permeability and retention effect of nanomaterials may induce the accumulation of radioisotopes in tumor tissue, making nanomaterials promising for cancer treatment [27,28].

This paper presents a comprehensive review of existing works on radiolabeled functional nanomaterials for cancer therapy. The first section focuses on synthetic aspects for incorporating radionuclides (α-/β-emitters) into nanomaterials. Next, recent studies on therapeutic applications of functional probes, such as targeted therapy and theranostic approaches in animal models, are discussed. Various nanomaterials investigated thus far and biomedical results of these studies are briefly described. Finally, future prospects and the emerging possibility of therapeutic radiolabeled nanomaterials are discussed.

## 2. Radiolabeling Methods

The stability of the therapeutic agent is a key factor in the successful implementation of radiolabeled nanomaterials for cancer therapy. Radionuclides should remain stable in nanomaterials, and the nanomaterials should maintain their integrity in biological environments and not be toxic to living subjects. Otherwise, released radioisotopes or nanomaterials could have significant adverse effects on normal tissues. Therefore, stable incorporation of radioisotopes is crucial for nanomaterial-mediated anticancer therapy. Radiolabeling can be performed using four different methods (Figure 2). The first method is the use of a chelating agent. Chelation is generally quite efficient for labeling α- and β-ray-emitting metal radioisotopes (e.g., ^67^Cu, ^90^Y, ^177^Lu, and ^225^Ac) on nanomaterials. However, an α-emitter and its daughters can be released from the chelator owing to the high recoil energy upon alpha decay, and they could cause considerable harm [29,30]. To deal with recoil, several nanocarriers—such as liposomes, carbon nanotubes, and multilayered inorganic particles—that can immobilize daughter radionuclides have been applied to prepare radiolabeled materials. In some studies, therapeutic radioisotopes were simply incorporated into nanoparticles comprising nonradioactive atoms (e.g., ^198^Au in gold nanoparticles, ^166^Ho in holmium nanoparticles). These methods were used for stable encapsulation of radionuclides. Alternatively, direct sorption of α-emitters (e.g., ^223^Ra and ^211^At) onto the surface of or inside nanomaterials was investigated by using the specific affinity of radioactive elements toward nanomaterials. Finally, radioactive iodine (^131^I), a commonly used β-emitter in clinical applications, can be attached to a phenol group on nanomaterials by oxidation followed by electrophilic substitution. The labeling method used ultimately depends on the research materials and purpose.

## 3. Beta-ray-Emitting Radioisotopes

A few β-ray-emitting radioisotopes, owing to their good availability and relatively low production cost, have been used for preparing radiolabeled nanomaterials for efficient cancer therapy. In general, β-particles have mean penetration depth of sub-millimeter range, and therefore, they can penetrate multiple tumor cells and show therapeutic effects. Table 1 shows the physical properties of commonly used β-ray-emitting radioisotopes.

### 3.1. Radioactive Iodine (^131^I)

^131^I (*t*_1/2_ = 8.02 days, E_β_ of 0.607 MeV) is one of the most frequently used β-particle-emitting radioisotopes in nuclear medicine. For several decades, radioactive sodium iodide ([^131^I]NaI) has been used for treating thyroid cancer [31,32]. ^131^I-labeled metaiodobenzylguanidine (^131^I-MIBG), another important radiopharmaceutical, is used for treating neuroblastoma and pheochromocytoma [33]. Owing to the increased demand for ^131^I in clinical practice, it is produced in large quantities and is normally available commercially. ^131^I also shows promise for molecular imaging studies using SPECT because it produces γ-ray emissions during decay. Radiolabeling of ^131^I in molecules typically involves electrophilic aromatic substitution of phenolic or trialkylstannylated substrates using an oxidant (e.g., chloramine T and iodogen); this normally affords high radiochemical yields. However, the deiodination of the resulting labeled product can often cause undesired accumulation of radionuclides in some organs such as the thyroid and stomach [34,35,36,37,38]. Therefore, it is essential to fabricate stable and multifunctional nanocarriers for efficient delivery of radioactive iodine to target tumor tissues. Table 2 shows ^131^I-labeled nanomaterials and their applications in vivo.

Several studies have investigated radioactive-iodine-labeled functional nanomaterials for cancer treatment. Liu et al. used albumin nanoparticles containing paclitaxel (PTX), a potent chemotherapeutic drug [40]. They assembled PTX with ^131^I-labeled human serum albumin (^131^I-HSA) to produce ^131^I-HSA-PTX nanoparticles. This material showed prolonged blood circulation time, specific tumor uptake, and high intratumor penetration ability. The combined therapeutic effects (chemo- and radiotherapy) of ^131^I-HSA-PTX were found to be highly effective in the 4T1 cancer xenograft model compared to radiotherapy- and chemotherapy-alone groups. In addition, nuclear images obtained using a γ-camera clearly revealed the in vivo behavior of nanoparticles and tumor localization. PTX and ^131^I-labeled copper sulfide nanoparticle (CuS NP)-loaded microspheres were also investigated for combination treatment of cancer cells [41]. The multifunctional therapeutic agent demonstrated highly effective combined photothermal (CuS NPs), chemo- (PTX), and radiotherapy (^131^I) of W256/B tumors in Sprague-Dawley (SD) rats. This material allowed noninvasive multimodal SPECT/photoacoustic imaging and showed the biodistribution of the injected agent and quantitative information about therapeutic effects. Nanosized reduced graphene oxide (RGO) has also been used for combination therapy [44]. Oxidized radioactive iodine could simply be incorporated into RGO to obtain the radiolabeled therapeutic agent (^131^I-RGO). RGO showed strong near-infrared absorbance and induced effective photothermal heating of tumor cells. The photothermal and radiotherapeutic effect of ^131^I-RGO resulted in effective elimination of tumors in an animal model. Liu et al. used polypyrrole (PPy) nanomaterial for tumor-targeted photothermal and radiotherapy of cancer [45]. To achieve enhanced tumor targeting ability, ^131^I-labeled transferrin (Tf), a tumor-targeting protein, was used as a capping agent of PPy. The resulting ^131^I-incorporated PPy showed enhanced in vivo tumor uptake and excellent therapeutic efficacy in the U87MG tumor xenograft model.

Recently, a simple one-step procedure was reported for preparing tumor therapeutic particles [46]. In this study, an aqueous solution of [^131^I]NaI was added to a mixed solution containing silver nitrate and ascorbic acid at ambient temperature to obtain ^131^I-doped silver nanoparticles (AgNPs) with high radiolabeling yield (98%). Interestingly, encapsulated radioactive iodine was stably retained inside the silver nanomaterial and showed good in vitro stability in mouse serum. The biodistribution results showed high tumor uptake values post intravenous (35.4%) and intratumoral injection (63.8%). These results indicated that ^131^I-doped AgNPs were a promising radiopharmaceutical. Several other nanomaterials including ^131^I-labeled dendrimers and polymers were also investigated for cancer treatment in vivo [42,43,47,48,49,50]. The radiolabeling procedure and therapeutic applications are similar to those in the abovementioned reports.

### 3.2. β-ray-Emitting Radioactive Metals

In general, metal radioisotopes can be labeled with a target material by using well-established bifunctional chelators. However, in some cases, an encapsulation method was used for preparing radiolabeled nanocarriers (e.g., ^166^Ho, ^198^Au). Table 3 lists radiometal-labeled nanomaterials and their biological applications. Radiometal-incorporated materials have also been used for various combination treatments of cancers.

Ytterium-90 (^90^Y) emits high radiation energy (2.280 MeV), and its average and maximum penetration depth in soft tissue are ~2.5 mm and 11 mm, respectively. ^90^Y has a physical half-life of 2.67 days that allow various radiotherapy applications using cancer-targeting peptides, antibodies, and resin microspheres [74]. ^90^Y-labeled nanomaterials have been used in some preclinical studies. A lipid nanoparticle containing a chemotherapeutic drug (doxorubicin) was labeled with ^90^Y for therapy of carcinoma cell line CNE1 in an animal model [51]. After the radiolabeled lipid particles were functionalized with folic acid, they were used for combined chemo- and radiotherapy. In this study, tumor growth was suppressed significantly compared to that in control groups. Another organic nanoparticle comprising *N*-(2-hydroxypropyl) methacrylamide (HPMA) was labeled with ^90^Y using DOTA and then used for radiotherapy of a prostate tumor (DU145 cells) model [53]. This study used gold-nanorod-mediated hyperthermia, and the combined treatment provided much higher therapeutic efficacy. Inorganic magnetic nanoparticles were labeled with ^90^Y for bimodal photothermal and radiotherapy in animal models [54,55]. The carboxylic acid groups on the nanoparticle surface efficiently immobilized the radionuclide and the radiolabeled nanomaterials showed high in vitro and in vivo stability; this may enable further potential applications for cancer treatment.

Lutetium-177 (^177^Lu, *t*_1/2_ = 6.73 days) produces β-particles with E_β_ of 0.5 MeV. The emission of low-energy gamma photons enables SPECT imaging of cancer cells and the biological distribution of radiolabeled products [75]. Reilly et al. reported ^177^Lu-labeled gold nanoparticles (^177^Lu-AuNPs) modified with a monoclonal antibody (panitumumab) for targeted therapy of EGFR-positive breast cancer [76]. This nanoprobe effectively suppressed the survival rate of cancer cells. The nanoparticle-based radiation treatment was also used in an in vivo theranostic study [56]. Radiolabeled AuNPs were injected intratumorally in animals bearing breast cancer cells (MDA-MB-648). Gold-nanoseed-mediated brachytherapy was highly effective at arresting tumor growth and increasing the survival rate of animal models. This research group also reported that ^177^Lu-AuNPs conjugated with trastuzumab can successfully be applied to local radiation therapy of HER2-positive breast cancer (MDA-MB-361 cells) [57]. A similar strategy was used for preparing nanomaterials modified with tumor-targeting peptides. Tyr^3^-octreotate [58] and cyclic RGD [59] were attached to ^177^Lu-AuNPs, and these probes were used for efficient radiation treatment of C6 gliomas tumor and HeLa cells, respectively. ^177^Lu-labeled PAMAM dendrimers were prepared by using a bifunctional DOTA-like ligand with pyridine-*N*-oxide [60]. The radiolabeled organic particles showed high in vitro stability and reasonable excretion kinetics in healthy animals, suggesting their promise as a radiopharmaceutical.

Rhenium-188 (^188^Re, *t*_1/2_ = 17.0 h) produces β-particles with E_β_ of 2.12 MeV, which can be produced by a commercially available W-188/Re-188 generator [77]. The advantage of this generator is that it produces both β- and γ-radiation components (0.155 MeV); therefore, the behavior of the labeled pharmaceutical can be monitored using SPECT. As the coordination chemistry of technetium (Tc) and rhenium (Re) is similar, conventionally used chelators for ^99m^Tc can be applied to the labeling of ^188^Re [78]. The lipophilic ligand dithiobenzoate coordinated ^188^Re complex was loaded in the core of lipid nanoparticles, and 12G5 antibody was conjugated to this probe for targeting U87MG cells in vivo [61]. Tumors were seen to regress significantly with radiotherapy and the ^188^Re-loaded nanocarrier enhanced the survival rate of the disease models. Similar radiolabeled lipid nanocapsules were also used for treating a human glioblastoma model [79]. Ting et al. reported ^188^Re and doxorubicin incorporated liposomes for bimodal therapy of colorectal adenocarcinoma (HT-29 cells) [62]. Although this probe has no targeting molecules, it showed high tumor/background ratio. Furthermore, SPECT images clearly visualized tumor cells and the biodistribution of the therapeutic agent. Similar radiolabeled liposomes have been evaluated in different cancer models [80,81]. Combined cancer therapy was also investigated by using ^188^Re-labeled magnetic nanoparticles (modified by human serum albumin) [63,82]. The chemotherapeutic agent Cisplatin was loaded to this probe and folic acid was used as a targeting ligand for treating human ovarian cancer. Trimodal (chemotherapy, radiotherapy, and hyperthermia) tumor treatment significantly increased the apoptotic rates of SKOV3 cells in animal models compared to control groups.

Gold-198 (^198^Au, *t*_1/2_ = 2.70 days) has beta decay energy of 0.960 MeV. Its average penetration depth in tissue is sufficient to provide therapeutic effects to destroy tumor cells. ^198^Au, owing to its favorable physicochemical properties, has attracted interest for the radiotherapy of several cancer cells [83]. ^198^Au radionuclide can be produced by the neutron irradiation of commercially available natural gold (^197^Au). In 2010, Jung et al. reported silica-coated AuNPs using neutrons in a nuclear reactor to convert non-radioactive gold to the ^198^Au radioisotope [65]. The structural integrity and stability of these core–shell particles were not affected under radiation, indicating that the product can be a useful radiotracer. Radioactive chloroauric acid (H^198^AuCl_4_) was also produced by the neutron irradiation of a gold source. The radioactive nanomaterial was synthesized by mixing H^198^AuCl_4_ and a reducing agent in the presence of a carrier (nonradioactive) NaAuCl_4_ solution to obtain ^198^Au-incorporated radioactive nanoparticles. Katti et al. developed a series of useful radioactive AuNPs functionalized by tumor-targeting agents including small molecules [66,67], peptide [68], and gum arabic glycoprotein [69,84,85]. These nanoparticles showed potent therapeutic efficacy for treating prostate tumor. For example, epigallocatechin-gallate-modified radioactive AuNPs significantly inhibited PC-3 xenograft tumor growth (80% reduction of tumor volume after 28 days of administration) in vivo compared to control groups [66].

Other β-emitting metal radioisotopes including ^64^Cu [70,86] and ^67^Ga [87] have also been used for synthesizing functional nanomaterials for in vivo applications. The therapeutic effect of these radiolabeled probes was evaluated in tumor xenografts. Notably, radioactive holmium (^166^Ho) was incorporated into several materials such as chitosan complex [88], acetylacetone nanoparticle [71,72], and iron garnet nanoparticle [73] to develop new therapeutic agents.

## 4. Alpha-ray-Emitting Radioisotopes

For several decades, α-emitters have been investigated for preclinical and clinical applications (Table 4). The key advantages of these radionuclides over β-emitters are the high levels of linear energy transfer and short penetration range in soft tissue (50–100 μm) [89]. Energy deposition occurs in a very small tissue volume with high relative biological effectiveness. Furthermore, the therapeutic effects of α-particles do not depend on hypoxia or cell cycles, making them more attractive for radionuclide therapy [90]. In particular, α-particle therapy requires highly stable attachment to carrier molecules to minimize the nonspecific radiation of healthy tissues by released radionuclides. Table 4 shows the physical properties of some α-emitters investigated in biomedical studies. Among these, short-half-life radioisotopes such as ^212^Bi (*t*_1/2_ = 60 min), ^213^Bi (*t*_1/2_ = 46 min), and ^226^Th (*t*_1/2_ = 30 min) are not suitable for cancer treatment. In recent years, ^211^At, ^223^Ra, and ^225^Ac were primarily used for preparing anticancer nanomaterials (Table 5).

Astatine-211 (^211^At) is a halogen radionuclide with a long enough half-life (7.2 h) to conduct radiochemical procedures. Its α-particle emission is associated with 100% of ^211^At decays, and ^211^At activity distributions can be quantified by SPECT because the decay branch involves electron capture decay (^211^At → ^211^Po) [112]. Because of its favorable physical properties, ^211^At is considered highly useful for cancer treatment. Various ^211^At-labeled antibodies have long been investigated for targeted alpha therapy (TAT) [113]. However, in vivo deastatination of ^211^At-labeled molecules, a notable problem, resulted in accumulation of free radioisotopes in specific organs including the thyroid. A theoretical study revealed that the C-At bond is not stable in the presence of oxidants and is weaker than its corresponding C-I bond [114]. Therefore, such dehalogenation should be considered for realizing effective therapeutic strategies. In 2017, ^211^At-labeled AuNPs were reported for TAT of cancer cells [91]. Iodine anion is known to have high affinity on the surface of AuNPs [115,116,117,118]. Considering this unique chemical property of halogen species, astatine, the heaviest halogen atom, is assumed to form stronger bonds with AuNPs than iodide ions. Simply mixing ^211^At with peptide-conjugated AuNPs under ambient conditions induced rapid chemisorption of radionuclides on the surface of nanomaterials. ^211^At-labeled AuNPs were quite stable in biological fluids and showed a potent cytotoxic effect in vitro on glioma cells. Recently, the same research group reported trastuzumab-modified AuNPs labeled with ^211^At (^211^At-AuNP-trastuzumab) for local treatment of breast cancer. ^211^At-AuNP-trastuzumab was effectively internalized and deposited near the nucleus of SKOV-3 cancer cells, and it showed higher cytotoxicity than nontargeted nanoprobes [92]. Silver-based nanomaterials [93,94] also show high affinity for astatine, and such ^211^At-labeled products were prepared by a sorption method. In another approach, ^211^AtCl_3_ was stably trapped inside single-walled carbon nanotubes (SWCNTs) by noncovalent interaction (van der Waals forces), suggesting that carbon-based nanomaterials can also be used as ^211^At carriers [95].

Actinium-225 (^225^Ac, *t*_1/2_ = 10 days) is a parent α-particle emitter in a decay scheme and produces a series of daughter α-particles. ^225^Ac-labeled molecules have shown high efficacy for radiotherapy of cancers in basic research and clinical applications [119]. In particular, the development of ^225^Ac-labeled radiopharmaceuticals has attracted increasing attention because of recent impressive results for prostate-specific membrane antigen (PSMA)-targeting enzyme inhibitors. The ^225^Ac-labeled small-molecule PSMA-targeting ligand showed better therapeutic effects than the corresponding β-emitter-labeled tracers [120,121,122]. As mentioned above, an α-emitter and its daughter radionuclides should remain stable at the target site for specific treatment of cancer cells in vivo. The use of well-known metal chelating agents such as DOTA, DTPA, and their analogs provided an efficient procedure for ^225^Ac-labeling. However, the release of daughter radionuclides such as ^211^Fr and ^213^Bi from the chelator can pose severe and nonspecific toxic effects in living systems [123,124]. Among various delivery systems, lanthanide phosphate (LaPO_4_) was investigated to retain sequestered daughters in the nanomaterials [96]. In this study, ^225^Ac-doped LaPO_4_ nanoparticles were synthesized by a simple fabrication process, and the surface of this material was conjugated with a monoclonal antibody for targeting in the lungs. The biodistribution study revealed that >80% of ^213^Bi is detected within the nanomaterial at the target region in animals after 120 h of administration, and these results were visualized by SPECT/CT. Similar multilayered LaPO_4_ materials incorporating ^225^Ac were prepared to develop efficient α-emitter-doped therapeutic agents. This material can successfully sequester the daughter ^213^Bi, and it showed significant reduction of EMT-6 tumor cells in the lungs [97,98]. Several studies have investigated the efficient encapsulation of α-particles in liposome nanoparticles. In 2004, Sofou et al. investigated several ^225^Ac-entrapped engineered liposomes to develop potent therapeutic agents [99]. In this study, zwitterionic and larger liposomes showed better retention (>88% over 30 days) of daughter radionuclides. The same research group also reported ^225^Ac-containing liposomes that were further conjugated with targeting molecules including monoclonal antibodies and aptamer for TAT [101,102,103]. For example, J591-antibody-conjugated liposome showed specific targeting ability and potent cytotoxicity to PSMA-expressing tumors (human LNCap and rat Mat-Lu cells) [103]. Carbon nanomaterials are also being studied as therapeutic cargoes of ^225^Ac. In 2013, Scheinberg et al. reported an efficient strategy for pretargeted therapy of cancer cells [106]. An anti-CD20-MORF was injected initially to target tumor cells in an animal model, and then, ^225^Ac-labeled SWCNT modified with a morpholino oligonucleotide sequence complementary to the sequence of anti-CD20-MORF was administered. Improved therapeutic effects were obtained by this two-step strategy. Because of the rapid renal clearance of the untargeted radiolabeled SWCNT, radioisotope toxicity was mitigated in living subjects. Ruggiero et al. reported a functionalized SWCNT covalently attached with the tumor targeting antibody E4G10 and a bifunctional chelator [107]. The radiolabeled SWCNT was applied to a xenograft model of LS174T, a human colon adenocarcinoma, and showed shrinkage of tumor volume and improved median survival rates (30%) compared to controls (0%) at 30 days after administration of the cancer-targeted material. ^225^Ac encapsulation within a nanocarrier can be another method to prepare a stable therapeutic agent. Through sonication, ^225^Ac was efficiently loaded in SWCNT (>95%) in the presence of Gd^3+^ metal ions to afford ^225^Ac@Gadonanotubes (^225^Ac@CNTs) [108]. This material showed high stability in human serum, suggesting that ^225^Ac@CNTs can be a useful anticancer agent for delivering high concentrations of ^225^Ac.

Radium-223 (^223^Ra) has also attracted much interest as a promising therapeutic radionuclide. Unlike other α-emitters, ^223^Ra can simply be produced from the ^227^Ac(*t*_1/2_ = 21.8 years)/^223^Ra generator [125]. The release of daughter nuclides from ^223^Ra is not a serious concern compared to ^225^Ac because most of the radiation energy of daughter α-particles decays within a few minutes. Owing to its favorable properties and therapeutic efficacy, radium dichloride ([^223^Ra]RaCl_2_) received the first marketing approval by the U.S. Food and Drug Administration (FDA) for the treatment of bone metastases from prostate cancer [126]. However, Ra^2+^ does not form a stable complex with conventionally used bifunctional chelators, similar to most alkali earth metals [127]. This unsatisfactory stability of radium radionuclides hampered the application of ^223^Ra in TAT. To overcome these limitations and develop a new class of radiotherapeutic agents, several inorganic nanomaterials such as zeolites [109,110], iron oxide nanoparticles [111], and hydroxyapatite particles [112] have been used for stable incorporation of radium. ^223^Ra was efficiently labeled with these materials by a sorption mechanism. Bilewicz et al. reported that the sodium form of A-type nanozeolite (NaA) could strongly bind with radium cations and its decay products [109]. The radiolabeling of a cancer-targeting peptide-conjugated NaA was performed by exchanging Na^+^ for Ra^2+^. The conjugate showed good in vitro stability (>90% retention over 6 days) and high cytotoxic effect toward glioma cells.

## 5. Conclusions and Future Perspectives

Various radiolabeled nanomaterials are being investigated for cancer treatment. This article reviewed the preparation methods of these materials and their results in biological applications. Nanomaterials can be designed to possess high loading capacity of therapeutic radioisotopes and to incorporate multivalent cancer-targeting molecules. These functional carriers can control the pharmacokinetics of radionuclides, and they show superior therapeutic effects. Furthermore, various useful molecules such as chemotherapeutic drugs and contrast agents can be loaded in the same materials to conduct combination treatments and theranostic studies. One of the most significant challenges associated with radionuclide therapy is the nonspecific radiation arising in normal tissues from unbonded radioisotopes. Therefore, stable incorporation of α-/β-emitters in nanostructures and accurate delivery to target tumor sites is a prerequisite for effective cancer treatment and mitigation of toxicity.

As discussed previously, therapeutic β-emitting radioisotopes are found in various organic, inorganic, and hybrid forms of nanomaterials. In most cases, radiolabeling is conducted by using a bifunctional chelator (for metal radioisotopes) or by forming covalent bonds (for radioactive iodine). In particular, a few β-emitter-labeled nanomaterials have been used in multimodal therapy to overcome the limitation of single-mode therapy. Indeed, several combination therapy strategies showed remarkable synergistic effects. Moreover, some therapeutic radioisotopes such as ^131^I, ^177^Lu, and ^188^Re also emit γ-rays, and thus, their incorporation provides a useful theranostic platform. These successful results and the good availability of β-emitters may enable the development of versatile nanoplatforms for effective theranostic studies. Compared to β-emitting medical radioisotopes, only a few research groups have investigated α-particle-incorporated nanomaterials because of their higher production cost and lower availability (e.g., limited production facility). Because typically-used chelating agents or labeling procedures normally cannot sequester the α-emitter and its decay products, the encapsulation of radionuclides within nanostructures (e.g., lipid nanoparticles, multilayered materials) or adsorption methods (e.g., inorganic nanomaterials) was considered preferable for obtaining stable therapeutic agents. These methods can reduce the detachment of the α-emitter and increase the retention rate of daughter radionuclides at tumor sites. Although nanomaterials have shown promising results for cancer treatment, some important issues regarding long-term toxicity and efficient clearance of radiolabeled materials from the body should be addressed for the successful translation of this technique into clinical applications. Therefore, more studies should focus on synthesizing radiolabeled nanomaterials having better radiochemical stability and desirable pharmacokinetic and excretion profiles to minimize radiation exposure in normal tissues. The improvement of optimized radiochemical procedures and extensive validation of the efficacy and toxicity of radiolabeled functional nanomaterials in various animal disease models will lead to the development of useful therapeutic radiopharmaceuticals. Considering the current achievements as well as high potential of α- and β-emitters, the ongoing preclinical study will continue to play an important role in the advancement of therapeutic technology.

## Figures and Tables

**Figure 1 ijms-20-02323-f001:**
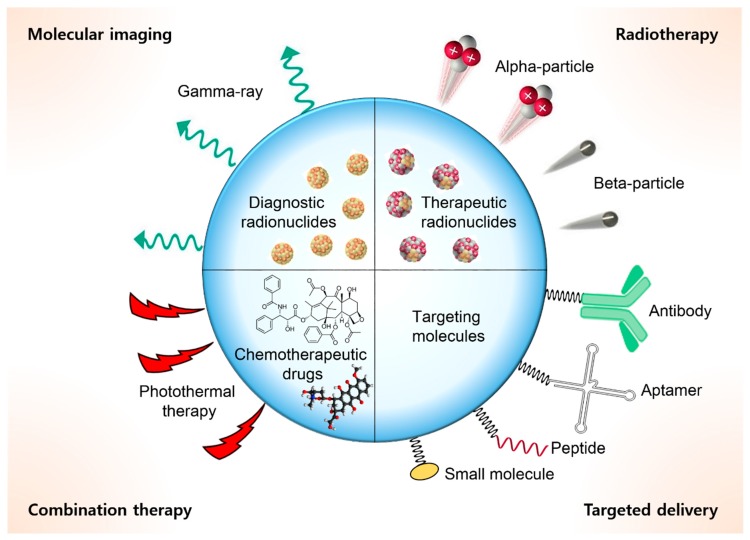
Multifunctional nanomaterials for therapeutic study.

**Figure 2 ijms-20-02323-f002:**
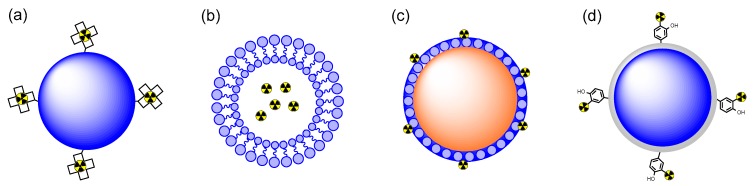
Incorporation of therapeutic radionuclides into the nanocarrier by (**a**) chelation, (**b**) entrapment, (**c**) sorption, and (**d**) covalent bonding.

**Table 1 ijms-20-02323-t001:** Physical properties of β-particles for medical applications

Radioisotope	Decay Product	Decay Half-Life	Mean Penetration Range in Tissue (mm)	Decay Energy (MeV_max_)
^131^I	^131^Xe	8.02 days	0.4	0.607
^67^Cu	^67^Zn	2.60 days	0.19	0.578
^90^Y	^90^Zr	2.67 days	2.5	2.280
^166^Ho	^166^Er	1.12 days	0.84	1.855
^177^Lu	^177^Hf	6.73 days	0.16	0.498
^186^Re	^186^Os	3.72 days	0.43	1.070
^188^Re	^188^Os	17.0 h	0.98	2.120
^198^Au	^198^Hg	2.70 days	0.38	0.960

**Table 2 ijms-20-02323-t002:** ^131^I-labeled functional nanomaterials

Nanomaterial	Labeling Method	Functions	Animal Model	Ref
Albumin nanosphere	Covalent bonding	Anti-AFP Mab (targeting), Doxorubicin (chemotherapy)	HepG2 cells (Balb/c mouse)	[39]
Albumin-paclitaxel nanoparticle	Covalent bonding	Paclitaxel (chemotherapy), SPECT imaging	4T1 cells (nude mouse)	[40]
CuS-nanoparticle-loaded microsphere	Covalent bonding	CuS nanoparticles (photothermal therapy), paclitaxel (chemotherapy), SPECT imaging	W256/B cells (SD rat)	[41]
PAMAM (G5)	Covalent bonding	Folic acid (targeting), SPECT imaging	C6-HFAR cells (Balb/c mouse)	[42]
PAMAM (G5)	Covalent bonding	Chlorotoxin (targeting), SPECT imaging	C6 cells (Balb/c mouse)	[43]
RGO ^a^	Covalent bonding	RGO (photothermal therapy), SPECT imaging	4T1 cells (Balb/c mouse)	[44]
Polypyrrole nanoparticle	Covalent bonding	Transferrin (targeting), polypyrrole (photothermal therapy), SPECT imaging	U87MG cells (nude mouse)	[45]
AgNP ^b^	Entrapment		WI-38 cells (Swiss albino mouse)	[46]
BSA-poly(ε-caprolactone) assembly	Covalent bonding	Cetuximab (targeting)	NCI-H1972 cells (Balb/c mouse)	[47]
BSA-poly(ε-caprolactone) assembly	Covalent bonding	RGD peptide (targeting), SPECT imaging	NCI-H460 cells (Athymic mouse)	[48]
Poly(HEMA-MAPA) nanoparticle	Covalent bonding	SPECT imaging	Healthy rats	[49]

^a^ Reduced graphene oxide; ^b^ Silver nanoparticles

**Table 3 ijms-20-02323-t003:** β-ray-emitting radiometal-labeled functional nanomaterials

Isotope	Nanomaterial	Labeling Method	Functions	Animal Model	Ref
^90^Y	Lipid nanoparticle	Chelation	Doxorubicin (chemotherapy), folate (targeting)	CNE-1 cells (BALB/c mouse)	[51]
	Lipid nanoparticle	Chelation	Anti-FLK Mab (targeting)	K1735-M2 and CT-26 cells (BALB/c mouse)	[52]
	HPMA copolymer nanoparticle	Chelation		DU-145 cells (nude mouse)	[53]
	Fe_3_O_4_ nanoparticle	Chelation	Fe_3_O_4_ nanoparticle (photothermal therapy)	Healthy Wistar rats	[54,55]
^177^Lu	AuNPs	Chelation	Panitumumab (targeting), SPECT imaging	MDA-MB-468 cells (CD-1 mouse)	[56]
	AuNPs	Chelation	Trastuzumab (targeting)	DMA-MB-361 cells (NOD/SCID mouse)	[57]
	AuNPs	Chelation	Tyr^3^-octreotate (targeting)	HeLa cells (in vitro)	[58]
	AuNPs	Chelation	Cyclic RGD peptide (targeting)	C6 gliomas cells (nude mouse)	[59]
	PAMAM (G1 and G4)	Chelation	-	Healthy Wistar rat	[60]
^188^Re	Lipid nanoparticle	Entrapment	12G5 mAB (targeting)	U87MG cells (SCID mouse)	[61]
	Lipid nanoparticle	Entrapment	Doxorubicin (chemotherapy), SPECT imaging	HT-29 cells (nude mouse)	[62]
	Magnetic nanoparticle	Chelation	Folic acid (targeting), cisplatin (chemotherapy), magnetic nanoparticle (photothermal therapy)	SKOV3 cells (BALB/c mouse)	[63]
	PAMAM (G5)	Chelation	Folic acid (targeting)	-	[64]
^198^Au	SiO_2_ ^a^ AuNPs	Entrapment	-	-	[65]
	AuNPs	Entrapment	Epigallocatechin-gallate (targeting)	PC-3 cells (SCID mouse)	[66]
	AuNPs	Entrapment	Mangiferin (targeting)	PC-3 cells (SCID mouse)	[67]
	AuNPs	Entrapment	BBN peptide (targeting)	PC-3 cells (SCID mouse)	[68]
	AuNPs	Entrapment	Gum Arabic glycoprotein (targeting)	PC-3 cells (SCID mouse)	[69]
^64^Cu	CuS nanoparticle	Entrapment	CuS nanoparticles (photothermal therapy), PET imaging	BT-474 cells (nude mouse)	[70]
^166^Ho	Holmium acetylacetone nanoparticle	Chelation	-	-	[71,72]
	Holmium iron garnet nanoparticle	Entrapment	Radiotherapeutic bandages	-	[73]

^a^ Gold nanoparticles.

**Table 4 ijms-20-02323-t004:** Physical properties of α-emitters for biomedical applications

Radioisotope	Alpha Decay	Decay Half-Life	Decay Energy (MeV_max_)
^211^At	^211^At → ^207^Bi	7.2 h	5.87
	^211^Po → ^207^Pb	0.516 s	7.45
^212^Bi	^212^Bi → ^208^Tl	61 min	5.87
	^212^Po → ^208^Pb	0.3 μs	8.79
^213^Bi	^213^Bi → ^209^Tl	46 min	6.05
	^213^Po → ^209^Pb	4.2 μs	8.38
^223^Ra	^223^Ra → ^219^Rn	11.43 days	5.78
	^219^Ra → ^215^Po	3.96 s	6.88
	^215^Po → ^211^Pb	1.78 ms	7.53
	^211^Bi → ^207^Tl	2.14 min	6.62
	^211^Po → ^207^Pb	516 ms	7.59
^225^Ac	^225^Ac → ^221^Fr	10.0 days	5.83
	^221^Fr → ^217^At	4.8 min	6.34
	^217^At → ^213^Bi	32.3 ms	7.07
	^213^Bi → ^209^Tl	45.6 min	6.05
	^213^Po → ^209^Pb	4.2 μs	8.38

**Table 5 ijms-20-02323-t005:** Representative studies on α-particle therapy using functional nanomaterials

Isotope	Nanomaterial	Labeling Method	Functions	Animal Model	Ref
^211^At	AuNPs	Sorption	Substance P (5–11) (targeting)	T98G cells (in vitro)	[91]
	AuNPs	Sorption	Trastuzumab (targeting)	SKOV-3 cells (in vitro)	[92]
	AgNPs	Sorption	-	-	[93]
	Ag/TiO_2_ nanoparticles	Sorption	-	-	[94]
	SWCNT ^a^	Entrapment	-	-	[95]
^225^Ac	LaPO_4_ nanoparticle	Entrapment	mAb 201b antibody (targeting), SPECT imaging	Healthy BALB/c mouse	[96]
	Multilayered LnPO_4_ nanoparticle	Entrapment	mAb 201b antibody (targeting), SPECT imaging	Healthy BALB/c mouse	[97]
				EMT-6 cells (BALB/c mouse)	[98]
	Liposome	Entrapment	-	-	[99]
	Liposome	Entrapment	Folate conjugated F(ab)’ (targeting)	OvCar-3 cells (in vitro)	[100]
	Liposome	Entrapment	Trastuzumab (targeting)	SKOV3-NMP2 cells (in vitro)	[101,102]
	Liposome	Entrapment	J591 antibody and A10 PSMA aptamer (targeting)	LNCap and Mat-Lu cells (in vitro)	[103]
	Polymersome	Entrapment	-	U87 cells (in vitro)	[104]
	TiO_2_ nanoparticle	Sorption	Substrate P (5–11) (targeting)	T98G cells (in vitro)	[105]
	SWCNT	Chelation	Morpholino oligonucleotide sequence (targeting)	LS174T cells (BALB/c mice)	[106]
	SWCNT	Chelation	Antibody E4G10 (targeting)	LS174T cells (BALB/c mice)	[107]
	SWCNT	Entrapment			[108]
^223^Ra	Nanozeolite	Sorption	Substance P (5–11) (targeting)	T98G cells (in vitro)	[109,110]
	Fe_3_O_4_ nanoparticle	Sorption	-	-	[111]
	Hydroxyapatite particles	Sorption	-	-	[112]

^a^ Single-walled carbon nanotube.

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
