# Peer review of "Review of Therapeutic Applications of Radiolabeled Functional Nanomaterials"

_ijms, 2019, doi:10.3390/ijms20092323_

Round 1
Reviewer 1 Report
This is very good prepared review on application of radioactive nanoparticles in medicine. As I have noticed, the review contains all relevant works on this subject. The publications were also well commented. Only English can be slightly improved.
Author Response
Point. This is very good prepared review on application of radioactive nanoparticles in medicine. As I have noticed, the review contains all relevant works on this subject. The publications were also well commented. Only English can be slightly improved.
Answer: I'd appreciate your favorable evaluation on my submitted manuscript. I have reviewed the manuscript to improve English style.
Reviewer 2 Report
The article written by J. Jeon provides a comprehensive review on the therapeutic applications of radioisotope-labelled nanomaterials to cancer treatment. This manuscript introduces various up to date studies on the biomedical use of various functional materials. In detail, the tables and figures shows the systematic summary on the synthesis, radiolabelling, applications including combination therapy, nuclear imaging and their efficacy in a few tumor xenograft animal models. Because this paper focuses on the novel use of nanomaterials for cancer therapy which is an active area of research, it should be of interest to readers in this area who seek a review of up to date use of nanomaterials for cancer therapy. In addition, this article could help acquaint researchers with existing approaches and suggest new potential research directions. Therefore, I suggest the publication of this review in IJMS after some minor revisions of the manuscript.
1. Please check the physical properties of alpha and beta emitters (table 1 and table 4). I found some wrong information on the table 1 including penetration range of 198Au, 131I and also check maximum decay energy of 131I (0.607 MeV is correct?).
2. In table 3, 77Lu should be changed to 177Lu.
3. In table 4, 212Pb should be changed to 212Bi.
4. It can be better if the author add several references about organic and inorganic nanomaterials for applications in preclinical studies.
Ex)
Silver Nanoparticles: Synthesis and Application for Nanomedicine, Int. J. Mol. Sci. 2019, 20, 865
Ultrasensitive NIR-SERRS Probes with Multiplexed Ratiometric Quantification for In Vivo Antibody Leads Validation, Advanced Healthcare Materials, 2018, 7(4):1700870.
Enzyme-catalyzed Ag Growth on Au Nanoparticle-assembled Structure for Highly Sensitive Colorimetric Immunoassay, Sci Rep., 8(1), 6290
Author Response
I'd appreciate your favorable evaluation on my submitted manuscript.
Point 1. Please check the physical properties of alpha and beta emitters (table 1 and table 4). I found some wrong information on the table 1 including penetration range of 198Au, 131I and also check maximum decay energy of 131I (0.607 MeV is correct?).
Answer 1. I have checked the table 1 and revised some of wrong information of physical properties of radioisotopes (e.g. mean penetration range of I-131, Au-198). For I-131, it decays mostly by beta-emission of 0.607 MeV (90%).
Point 2 and 3. In table 3, 77Lu should be changed to 177Lu. In table 4, 212Pb should be changed to 212Bi.
Answer 2. I have revised accordingly.
Point 4. It can be better if the author add several references about organic and inorganic nanomaterials for applications in preclinical studies.
Answer 4. I added three more references in the manuscript. Thanks for good suggestion.